# Real-Time Monitoring of Yogurt Fermentation Process by Aquaphotomics Near-Infrared Spectroscopy

**DOI:** 10.3390/s21010177

**Published:** 2020-12-29

**Authors:** Jelena Muncan, Kyoko Tei, Roumiana Tsenkova

**Affiliations:** Biomeasurement Technology Laboratory, Graduate School of Agricultural Science, Kobe University, Kobe 657-8501, Japan; kyungja0720@yahoo.co.jp

**Keywords:** yogurt, fermentation, real-time monitoring, near-infrared spectroscopy, aquaphotomics, lactic acid bacteria, moving window principal component analysis, water

## Abstract

Automated quality control could have a substantial economic impact on the dairy industry. At present, monitoring of yogurt production is performed by sampling for microbiological and physicochemical measurements. In this study, Near-Infrared Spectroscopy (NIRS) is proposed for non-invasive automated control of yogurt production and better understanding of lactic acid bacteria (LAB) fermentation. UHT (ultra-high temperature) sterilized milk was inoculated with Bulgarian yogurt and placed into a quartz cuvette (1 mm pathlength) and test-tubes. Yogurt absorbance spectra (830–2500 nm) were acquired every 15 min, and pH, in the respective test-tubes, was measured every 30 min, during 8 h of fermentation. Spectral data showed substantial baseline and slope changes with acidification. These variations corresponded to respective features of the microbiological growth curve showing water structural changes, protein denaturation, and coagulation of milk. Moving Window Principal Component Analysis (MWPCA) was applied in the spectral range of 954–1880 nm to detect absorbance bands where most variations in the loading curves were caused by LAB fermentation. Characteristic wavelength regions related to the observed physical and multiple chemical changes were identified. The results proved that NIRS is a valuable tool for real-time monitoring and better understanding of the yogurt fermentation process.

## 1. Introduction

Fermented dairy products have a long history of production. They are traditionally considered to have many health benefits and, as such, are of natural interest to the dairy industry. These functional dairy food products have been increasingly used in daily life, and the rise in popularity of consumption on a world-wide scale can be attributed to the increasing evidence of many positive health effects [1,2,3,4].

One of the most popular functional dairy foods is yogurt, a product of lactic fermentation of milk [5,6]. The majority of yogurt production is performed using bovine milk [7]. Lactic acid fermentation is a process of conversion of milk sugar-lactose (C_12_H_22_O_11_) into the lactic acid (C_3_H_6_O_3_) by the use of lactic acid bacteria (LAB), thereby resulting in a decrease of the pH value [8]. The two homofermentative bacterial strains are usually used for this purpose-*Streptococcus salivarius* subsp. *thermophiles* and *Lactobacillus delbrueckii* subsp. *bulgaricus* [9]. The acidification is a key mechanism responsible for coagulation during yogurt fermentation [9]. The thick structure of the coagulated product-yogurt is due to the casein proteins forming the gel matrix.

Caseins are a family of phosphoproteins commonly present in milk, in the form of suspended particles—casein micelles—held together by colloidal calcium phosphate, which binds together numerous submicelles. Acidification of milk, i.e., lowering the pH leads to a dissolution of colloidal calcium phosphate and release of casein content from the micelles. During this process, three phases could be observed depending on the pH [9]. First, at the beginning of acidification, when the pH value is reduced from 6.7 to 6.0, only a small amount of colloidal calcium phosphate is dissolved, so the structural changes of the micelles are limited. In the second phase, when the pH is decreased to 5.0 the colloidal calcium phosphate is completely dissolved, and lastly, when the pH decreases toward the isoelectric point of the casein (pH 4.6), the casein micelles aggregate and form the gel matrix of yogurt. Usually, the endpoint of yogurt fermentation is decided when the pH reaches 4.6, which is when all the casein is in solubilized form, and the gel matrix of the yogurt is created.

Since the entire process largely depends on the pH, this property of milk is one of the indicators mostly used to monitor and evaluate the progress [10,11] and has been proposed for continuous, online control of yogurt fermentation [9]. Despite the easy implementation and very low cost of pH measurement techniques, in industrial settings, the pH is still usually measured discontinuously, using pH probes, titration methods, or pH indicator solutions [12]. Titration methods and pH indicators require sampling and time-consuming laboratory analysis; hence they are not suitable for continuous control. The continuous pH measurements using probes are limited by the slow stabilization of the probes, the drift of the probes, deposition of protein material, and sanitation problems [9,11].

Improving the yogurt fermentation process by automatic monitoring and control could have a large impact on the dairy industry in the form of increased quality of the products and economic gains. One of the main problems to be addressed in this regard is achieving high consistency and reproducibility of the fermentation process [13], which will result in the consistent quality of the final product. The process of yogurt formation is intricate, and the progress is not uniform throughout the fermenter. There are far more variables that are connected to the final product quality than a single parameter such as pH.

At present, various forms of sampling for physical measurements, microbiological, and chemical examinations are required for the monitoring. Unfortunately, rapid, reliable, and robust monitoring techniques for this purpose are still lacking in the dairy industry [13] and are a subject of intensive research.

In recent years different techniques have been investigated and proposed for continuous fermentation monitoring, among which the ones with the highest potential are: electronic nose (EN), electronic tongue, and near-infrared spectroscopy (NIRS) in addition to standard bioreactor probes [11,13,14,15,16,17]. Also, a fusion technology, combining EN and NIRS into one online sensor, has been investigated for monitoring of yogurt fermentation [14]. While the EN data are used for the measurement of chemical parameters, the NIRS is used for the measurement of physical parameters of coagulation. Many other methods have been investigated, such as infrared spectroscopy [18], Terahertz time-domain- attenuated total reflection spectroscopy [10], acoustical measurements [19,20], measurements of dielectric properties [5] and electric impedance [21], nuclear magnetic resonance spectroscopy [22,23,24], image analysis of laser diffraction patterns [25], and fluorescence spectroscopy [7].

NIRS has a long history of successful application as a very suitable, non-destructive method for quantitative and qualitative analysis of milk and dairy products [26,27]. Online measurements by NIRS have been applied for the determination of milk constituents: fat, lactose, protein [28,29,30,31,32,33], fatty acids [34,35,36,37,38], and in process control of cheese production [39] and monitoring of yogurt fermentation [7,40,41]. In-line, real-time monitoring has been done for milk coagulation [42] and monitoring of milk components [33]. This method could also provide real-time, non-destructive pH monitoring, as one of the parameters of yogurt quality control, since there are studies reporting the successful pH measurements in different systems [43,44], and in yogurt and filmjölk as well [13].

The possibilities of NIRS are further extended owing to the development of aquaphotomics [45], a novel science and methodology which utilizes the near-infrared (NIR) spectral features of water in the aqueous and biosystems for indirect measurements of components of interest, as well as integrative analysis, using a water spectral pattern (WASP) as a biomarker connecting the state of the water molecular structure of the analyzed system directly to system’s functionality [46,47]. The introduction of aquaphotomics showed the possibility for much more complex evaluations, expanding the limits of NIRS to disease diagnosis and pathogen identification [26]. This integrative approach allowed the successful application of NIRS for complex problems such as disease diagnosis or prediction of different physiological states based on spectra of various biofluids (milk, blood, plasma, and urine) or tissues in plants and animals [48,49,50,51,52,53,54,55,56]. This progress is made owing to the better understanding of water as an intrinsic component of every aqueous or biological system in which the water molecular network is interconnected with all other components present in it, making the water behave as a cumulative sensor [47,57]. Aquaphotomics method has been used for various purposes so far in food quality monitoring, and also investigations were made in the microbiology field in relation to the production of yogurt [47]. Bacteria studies were concerned with the classification of bacterial strains and the selection of those which are better probiotic candidates [58,59,60]. The water spectral patterns (WASPs) [45,61] of different bacteria clearly classified them into three groups according to their probiotic ability to non-probiotic, moderate, and probiotic [59]. Another study determined that features of NIR spectra and corresponding water spectral patterns of bacteria originate mainly from the metabolites the cells produce, rather than cells themselves, which is a strong indicator that water spectral pattern is very sensitive to workings of bacteria metabolism and products released in the environment [62]. Additionally, aquaphotomics was shown to be a good method for screening of fermentation processes of different microorganisms [63,64].

The objective of this study was to use aquaphotomics NIRS to study yogurt fermentation, find which spectral features carry information about the progress of the process, and to obtain a better understanding of LAB growth and coagulation of yogurt. There are several studies already utilizing NIR and/or infrared spectra of milk and yogurt that extracted features related to the compositional changes and other properties during fermentation, showing the significance of water absorbance bands such as the second overtone, overtone of the combination band, first overtone of water and the fundamental band [65,66,67].

The listed studies provide a good rationale to apply aquaphotomics NIRS for capturing the spectral changes related to yogurt composition, indicative of both chemical and physical variations that occur during the fermentation process, while at the same time providing more information about the process itself.

## 2. Materials and Methods

### 2.1. Spectral Data Acquisition

UHT sterilized 500 mL milk (Oishii-gyunyu, Meiji Co., Ltd. Tokyo, Japan) was pre-warmed to a temperature of 38 °C in a microwave oven. The milk was inoculated with 10 mL of Bulgarian yogurt (Bulgaria yogurt, Meiji Co., Ltd., Tokyo, Japan) and stirred for 2 min at 1500 rpm using a magnetic stirrer. The inoculated milk was then placed into a quartz cuvette cell with 1 mm path length for the spectral measurement and distributed equally to 17 test-tubes containing 10 mL samples for the measurement of pH values, respectively.

In order to explore the possibility of monitoring the fermentation process by NIRS, FT-NIR Multi-Purpose Analyzer-MPA (Bruker Optics, Ettlingen, Germany) instrument was chosen since it enables automatic measurements and removes the need for personnel. The spectrometer was equipped with a TE-InGaAs detector and temperature controller connected to the cuvette cell holder to provide the constant temperature. The optimum growth temperature for LAB is between 35 °C–42 °C [68,69,70,71,72], according to some studies, even until 45 °C [73]. The fermentation temperature 38 °C was chosen as approximately in the middle of this range, towards lower values to have a longer fermentation time for monitoring and because it enables better sensorial properties (in terms of the denser protein matrix, and more viscous, smoother, and slimy texture) [70,71,74,75,76]. The sample in the cuvette cell was fermented in the spectrometer at the temperature of 38 °C, and absorbance spectra of the same sample were acquired every 15 min, during the period of 8 h, continuously. All spectral data in the range of 12000–4000 cm^−1^ (~830–2500 nm) were obtained by 32 co-added scans with the 16 cm^−1^ spectral resolution. Air was measured as the background.

The samples in the test-tubes were fermented in the incubator (Sanyo Electric Co., Ltd., Moriguchi City, Osaka, Japan) at the same 38 °C temperature. The pH measurements were performed every 30 min during the fermentation, and at every measurement time point, the new tube was opened and pH measured. The pH meter, B211 (Horiba Ltd., Kyoto, Japan) was used for measurements.

### 2.2. Spectral Data Analysis

The spectral data were acquired using OPUS software (Version 3.1; Bruker Optik GmbH, Ettlingen, Germany) and analyzed using MATLAB (version 6.5; The Math Works Inc., Natick, MA, USA).

The spectral data showed noise at the spectral edges, and especially at wavelengths longer than 1900 nm due to the performance of the detector and the high absorbance of the sample. For this reason, subsequent analysis was carried out using the truncated 950–1880 nm region. The raw data were subjected to Savitzky–Golay smoothing [77] (second order polynomial filter, 25 points) before further analysis.

Moving window principal component analysis (MWPCA) was applied to elucidate details in physical and chemical changes while monitoring the process of the fermentation. MWPCA was proposed as a very suitable tool for monitoring and improving the performance of processes [78]. A schematic representation of the MWPCA algorithm is shown in Figure 1.

MWPCA is based on the concept of principal component analysis (PCA) [79]. This analysis scans and monitors the changes in the direction of each principal component over time. The time-window with *h_s_* size moves over the whole region to select submatrix X*_i_* of spectral data X*_m_*
_× *n*_. The window starts at the *i*th spectral channel and ends at the (*i* + *h_s_* − 1)th spectral channel. Each principal component is calculated for each X*_i_* of X. In the present work, first principal component is used in the calculation of the dissimilarity index and in *i*th submatrix X*_i_*. It is abbreviated as *p*(*i*). In order to detect a change of *p*(*i*)s, *p*(0) is defined as reference *p*(*i*)s, and the differences between *p*(*i*)*s* and *p*(0) are used as indexes for monitoring. The following index, dissimilarity index *A_i_* is used for evaluating a change of *p*(*i*).
$$Ai=1-\left|p{\left(i\right)}^{t}p(0)\right|$$

The dissimilarity index is based on the inner product of *p*(*i*) to *p*(0) $p{\left(i\right)}^{t}$-transposed vector), i.e., the angle of *p*(*i*) to *p*(0). When the *i*th first principal component, *p*(*i*) is equivalent to its reference, *p*(0), *A_i_* becomes zero. However, *A_i_* becomes one when *p*(*i*) is orthogonal to *p*(0).

Applying MWPCA was performed with a window size of five samples in the spectral range of 954–1880 nm. The window size was chosen to provide a small enough window to efficiently capture the changes in the fermentation process but large enough to not be too influenced by undesirable factors (noise, outliers, etc.) [80]. In order to find absorbance bands correlated with the chemical changes in the sample during fermentation, the variance of the loadings was also calculated.

## 3. Results

### 3.1. Monitoring of Physical Characteristics during Yogurt Fermentation

Original (raw) NIR absorbance spectra of yogurt acquired during fermentation in the cuvette cell are shown in Figure 2. The spectra closely resemble those collected during milk lactic acid fermentation and yogurt fermentation, which are already reported in the literature [13,67]—a trend of increasing absorbance with the progress of fermentation can be clearly observed. This increase is actually a result of baseline drift caused by scattering. Milk is a strong absorber of NIR light, but it also scatters light strongly owing to the colloidal protein dispersions, fat emulsion, and differences in the size of such particles [81]. The amount of light scattering would highly depend on the number and size of the scattering particles. The majority of scattering in milk is from the big fat globules, and less from smaller-sized casein micelles. However, the acid coagulation of milk using LAB causes an increase in the size of the casein micelles, which become larger than fat globules and can strongly vary in diameter (80–300 nm) [82]. The baseline changes, hence, were considered to be a result primarily from light scattering by casein micelles.

Other distinctive spectral features are two very dominant water absorbance bands around 1450 nm and 1910 nm due to the first overtone of the OH stretching vibrations and combination region of OH stretching and bending vibrations (combination band), respectively [83,84]. The smaller features noticeable around 975 nm and 1150 nm were also due to water, corresponding to the second overtone and the overtone of the combination band of water [83,84].

From the raw absorbance spectra, it can be observed that, in addition to the scattering effects, the main characteristics of the NIR spectra of yogurt during fermentation arise from water—the main component of milk (around 83%) [85], and consequently major absorber of NIR light.

In order to establish the relationship between the baseline effects and progress of fermentation, the values of pH measured in test-tubes during fermentation were plotted against time (Figure 3a) and compared to the changes of the baseline in spectra acquired at the same time points from a yogurt fermented in the cuvette cell. For this purpose, three wavelengths that showed minimum absorbance throughout the process 1050 nm, 1245 nm, and 1760 nm, were selected, and the absorbance at those bands plotted as a function of time, in the same manner as for the pH (Figure 3b). These wavelengths were selected based on the 0 values in the second derivative transformation of the spectra (data not shown) because at those points, the spectra were considered to be influenced only by the slope of the baseline.

The pH profile of yogurt during fermentation, as expected, showed three distinctive phases similar to the different stages of bacteria growth [11]: (1) lag phase (slow pH decrease), (2) logarithmic phase (rapid pH decrease), and (3) slow-down of acidification rate. The exact shape of the fermentation curve can be different depending on a great number of parameters, such as milk base, the type and amount of added ingredients, starter culture, the temperature of incubation and heat treatment of milk, but in general, pH profiles expressed as a function of incubation time could very well be described by the equation for bacterial growth [11]. The characteristic S-shape of the pH profile curve confirms the conversion of lactose to lactic acid and that the fermentation process is in progress [86].

The acidification of milk due to the bacteria growth and metabolic activity causes different occurrences at the molecular level and reorganization of the milk structure.

Between pH levels of 6.6 and 5.3, lowering milk pH caused by LAB conversion of lactose into lactic acid results in the isolation of calcium and phosphate from casein micelles, but their size (diameter) still remains approximately constant because a very small amount of colloidal calcium phosphate is dissolved in this phase [7]. During the decrease in pH to 5.0, the value drops rapidly, closely reflecting the major structural changes of casein. At values of pH between 5.2–5.3, precipitation of calcium phosphate occurs and subsequently causes denaturation and aggregation of casein micelles. Between pH levels of 5.3–4.6, these aggregates contract into smaller areas, and finally, individual casein particles are formed again, including fat globules and whey, larger than the original casein micelles and fat globules. They are different in character resulting from the loss of calcium phosphate [87], and usually, yogurt fermentation is terminated when the pH reaches 4.6, an isoelectric point of casein.

Examination of the absorbance time profiles at selected 1050 nm, 1245 nm, and 1760 nm wavelengths (Figure 3b) also showed three distinctive phases of changes in the baseline slope. The baseline slope increase started at pH 5.2–5.5 and paused at pH 4.6. This result showed substantial baseline slope changes with acidification of milk and values of pH. This slope change is not normally present in the spectra of milk, so it can be concluded that it was the outcome of fermentation. Relating these findings with the reorganization of milk structure on a molecular level, it could be concluded that these effects on the spectra were caused by the change in the shape of casein micelles and properties of the media as a result of increased acidity. In the process of acid production, the average size of casein particles is relatively constant in the pH range of 6.6 to 5.3, but when the pH is 5.3 to 4.6 or less, casein destabilization and particle aggregation occurred, and the micelles grew larger. The rapid change in the baseline occurred exactly in the range where the size of the micelles increased.

Hence, the changes in physical properties that occur from the fermentation of yogurt were the source of baseline variations. The baseline slope changes were also a result of the increase of overall water absorption because of the changed properties of the media. Denaturation of milk protein produces curd; in other words, the media changed into a gel from a liquid in the fermentation process; essentially, it is a process of phase transition. The media gelation simultaneously resulted in the scattering of light and caused the extension of optical path length, which has an effect on longer wavelengths. Thus, overall water absorption in the media increased, and it contributed to the baseline slope changed.

### 3.2. Monitoring Structural Changes during Yogurt Fermentation

The original data were subjected to Savitzky-Golay smoothing [77] before applying MWPCA with a window size of 5 samples in the spectral range of 954–1880 nm. Dissimilarity index *A_i_* showed changes as the time window moved (Figure 4). The *A_i_* index increased remarkably between 11th and 18th window points; in other words, the difference between the directions of the first principal components and the reference became larger in that period. In order to detect more detailed chemical information, the loadings were classified into three stages, according to the dissimilarity index (Figure 4), and subsequently, their variance with each stage was calculated and presented for stage two in Figure 5 and for stages one and three together in Figure 6. The duration of each stage was: (1) Stage one: 0–3 h 15 min, (2) Stage one: 3 h 15 min–4 h 30 min, and (3) Stage three: 4 h 30 min–8h. Note that each of the stages observed based on the Dissimilarity index plot (Figure 5) coincide with the observed phases in pH changes and baseline changes. In other words, the sigmoidal shape of the dissimilarity index also closely resembled the typical shape of a microorganism growth curve related to the fermentation of yogurt.

By examining the loading transition represented by MWPCA it was now possible to observe specific absorbance bands at which the absorbance changes as a result of various chemical changes the sample goes through during the fermentation process. By setting the window size, it was possible to identify chemical changes of the greatest influence on fermentation while various processes are occurring.

From the plots of the loading variances of stage two ranging from the eleventh to the seventeenth window position (Figure 5), it could be observed that the magnitude was far larger, by several orders, compared to the stage one ranging from first to the tenth window and stage three ranging from the eighteenth to twenty-ninth (Figure 6). This time window in stage two coincided with the time of rapid pH decrease and logarithmic growth phase of bacteria (Figure 3).

At stage two, loading variance is high in the regions above 1450 nm and below 1200 nm, while it is close to zero around 1300–1360 nm. The loading variance in stages one and three show the importance of more localized regions and bands. In stage one, distinctive bands appeared at around 970 nm, 1000 nm, 1090 nm, 1361 nm, 1670–1675 nm, and 1840 nm. Stage three showed high variance at 1420 nm and 1453 nm, slightly higher absorbance in the whole area around 1050 nm, and very high absorbance, which increased with wavelengths above 1500 nm.

Interpretation of the assignments of absorbance bands located at the above-found locations of highest variance in loadings of MWPCA during three found stages revealed the intricacies of the fermentation process during which milk is transformed into yogurt.

## 4. Discussion

Based on the above exploration, it can be concluded that the basic changes in the NIR spectra of milk during fermentation to produce yogurt originate from two sources—physical transformation, which is due to changes in light scattering properties of the particles changing in size and density, and chemical changes from mainly one highly absorbing component—water.

The main spectral regions where the change in absorbance takes place during the fermentation process were located around 970 nm, 1450 nm, and 1840 nm, as revealed by analysis of loadings variances of MWPCA. These regions can be attributed to the second overtone of OH stretching, the first overtone of the OH stretching, and the OH combination bands of water, respectively [88,89,90]. With water being the main component of both milk and yogurt, it was expected that the most intense absorption bands would be resulting from vibrations of water molecules. Depending on the stage of fermentation, loading variances showed marked differences in those regions.

In stage one, the bands 970 nm, 1361 nm, 1670–1675 nm, and 1840 nm underwent major changes. In the vicinity of the 970 nm band, there were several more peaks and troughs noticeable: around 988 nm (trough), 998 nm (peak), and a small shoulder-like feature around 1030 nm. The peak at 988 nm can be linked to the lactic acid absorption, bands at 970 nm and 998 nm represent water species with two and three hydrogen bonds, respectively, while peaks 1026–1032 nm are reported to be associated with casein–protein absorptions [91]. According to Saranwong and Kawano, this spectral region contains the information from both absorptions of milk compounds—lactic acid, lactose, urea, and casein, as well as water [91]. Bands located at around 1025 nm and 1030–1032 nm are also assigned as protein bands in other works [88,90,92]. However, Sasic and Ozaki attributed the band 996 nm to the second overtone of water interacting with protein [90], while Al-Quadiri et al., who monitored microbial spoilage of milk, considered the spectral variations at the second overtone of water region as indicative of proteolysis [92]. In another study, the light absorbance in the entire 600–1000 nm region, especially at 998 nm was found to be negatively correlated with pH [93]. Considering all this together, it is possible to conclude that this region shows the production of lactic acid and that proteolysis is in effect.

The absorbance band at 1361 nm is located in the area of the first overtone of water and can be assigned to proton hydrates/solvated hydronium ion [94,95] and/or water hydration shells [45]. In either case, it can be attributed to water involved with the hydration of charged structures. The bands of protonated water were shown to feature strongly in the spectra of different bacterial strains and, in addition to free water bands, change most during the LAB cultivation [60]. The study by Slavchev et al. revealed that prominent bands of protonated water had a high power of discrimination between low-pH resistant and non-resistant strains of LAB, hence connecting the protonated water with pH of the media and metabolites the specific strains produce [60]. Another study recently related protonated water to the self-organization of fatty acids in milk [38]. The micellar destructuration induced by acidification is similar for different types of acids (lactic acid included), in the terms that mainly pH is what governs the physicochemical changes. The changes in protonated water during the first stage of fermentation most probably reflect the destructuring of casein micelles. The acidification, in addition, also involves the protonation of organic and inorganic phosphate, citrate and carboxylic residues of caseins, while at the same time aqueous phase becomes less saturated in calcium phosphate due to the dissociation of this salt [96]. The extent of mineral and casein dissociation depends on pH, and at pH 5.2 calcium is partially, while inorganic phosphate totally solubilized. The casein molecules during acidification bind protons and become less negatively charged until they are neutral at around pH 4.6, their isoelectric point, when precipitation or gelation is at place [96].

Taken together, it most probably means that the changes at 1361 nm reflect both changes in solubilization of minerals and binding of protons by casein, which influence the changes in charge and destructuring of casein micelles, followed by the changes in the milk water fraction and solvation of minerals. After the first stage, this band was absent, which probably indicates that casein molecules are aggregated and their solubility minimal, as can be expected. Since the entire 1300–1400 nm region seems to be of importance for the first stage of fermentation, it should be noted that this region corresponds to the absorbance of weakly, or non-hydrogen bonded water, indicating the existence of a liquid phase, in other words, that milk is still more liquid-like, not coagulated.

One more feature of the first stage fermentation is a strong band centered around 1670–1675 nm, extending to almost 1900 nm, with an additional peak prominent around 1840 nm. This band could be attributed to the exopolysaccharides (EPS)—a product of LAB, whose absorbance bands are located within this region at 1683 nm, 1722 nm, and 1752 nm, and which influence the rheological parameters of fermentation product [97]. The creamy, smooth yogurt texture is associated with the bacteria that can produce EPS [98], like the one used in the study. The bands in this area can also be attributed to the absorbance of different water species; bonded OH of H_3_O^+^ stretch in small proton hydrates H^+^(H_2_O)_4_ at 1668 nm [95], and H^+^(H_2_O)_6_ at 1673–1674 nm [94,99]. The texture of yogurt is influenced by EPS acting like a viscosifying/gelling agent that interacts and binds the free water in the gel-like structure [97,98,100,101,102].

The entire 1300–1950 nm region is strongly influenced by water, which was presented by main absorption peaks around 1450 nm and 1900 nm. The 1900 nm absorption was a vibrational overtone combination of the O–H stretch and the H–O–H bend caused by molecular water, and the peaks at different wavelengths could be attributed to different binding states of water [103,104]. The band 1840 nm was in the past studies found to feature as an important variable for prediction of microbial mass (in soil, at 1842 nm) [105], sucrose content [106], pectin (polysaccharide) content and cellulose (polysaccharide), where it is attributed to the combination of OH stretching and CO bonds [107]. While it seems that this band is connected to microbes and polysaccharides, it was also found to be related to the water species present in minerals—either physically adsorbed on the surface of mineral grains, occupying specific lattice sites, or as a part of the crystal structure [108]. Since the water is a major component of yogurt, it is more likely that this absorption comes from the water-EPS interaction. It is reported that one of the major problems with EPS quantification is that the impossibility to remove bound water impedes an accurate determination [109]. Numerous studies reported that EPS production affects yogurt texture and physical properties [110]; however, a better understanding of the structure-function relationships in a yogurt matrix remains a major challenge [111]. What our results suggest is that the region above 1500 nm reflects an interaction and binding of water by EPS, which is actually the process that defines the water holding capacity of yogurt and its firmness. Since previously explained results showed active acidification and changes in the physicochemical environment of casein micelles, the EPS-water interaction shows successful triggering of the gelation of milk [112]. The viscosity, water-holding capacity, hardness, and microstructure (the size and density of interspaced voids) of yogurt are distinctly affected by the EPS quantity in yogurt [113].

Stage two of yogurt fermentation was characterized by very high loadings variance in the regions above 1450 nm and below 1200 nm, while almost none in the region 1300–1360 nm. This stage coincides with the rapid decrease in pH when denaturation and aggregation of casein micelles happen, and the gel matrix is formed. The high variance in both regions could be connected with the increase of hydrogen bonding and strongly bound water. The aquaphotomics studies researching the state of water in hydrogels connected the high absorbance at the region above 1500 nm to the quality of the polymer network of hydrogels and water binding capacity [114,115]. Also, significant bands associated with microorganisms were observed around 1520 nm and 1930 nm, which were linked to the protein content and with C=O bond, while the region from 1640 to 1786 nm to bacterial aromatic/heterocyclic groups, all of which were connected to microbial structures [116,117,118]. It is also interesting that the characteristic wavelengths for monitoring pH were distributed in 1460–1560 nm and 1800–1850 nm regions, respectively [119]. These reports show the huge informative value of the spectral features in this NIR region; however, since the water is still the major part of yogurt, it is likely that observed changes are actually attributable to water in interaction with other compounds reflecting their changes as well. The drop in variance at 1300–1360 nm could be attributed to the depletion of free water molecules, small protein hydrates, and weakly hydrogen-bonded species, which were being bonded into the formed gel matrix of the yogurt consisting of casein aggregates and EPS.

Stage three was characterized similarly to stage two with a high variance of loadings in the area above 1500 nm, showing the bound water in the gel matrix of yogurt. An interesting feature was the appearance of small bands at 1420 nm and 1453 nm. These were very close to the reported positions of bands at 1425 nm and 1455 nm, attributed to intermediate water structures (not free, not bound water molecules). Specifically, 1425 nm is associated with hydration water, water hydrating proteins [45,47,120,121], while 1455 nm is ascribable to adsorbed water [122], which is probably physically adsorbed at the pores of the yogurt microstructure, and interestingly, found to be connected to the inhibition of bacteria [123]. Previous works reported that the interplay between these two bands is strongly related to water activity [124]. The intensity and the direction of changes in absorbance at 1425 nm and 1455 nm upon changes in the water content were reported to be different for different systems [124,125], depending on the nature of the sample matrix (the pore sizes of the sample matrix, the evaporation rate, etc. [126]). Alternatively, a very close 1428 nm band may be related to the glucose molecules, the basis of complex polysaccharides [127], which indicates possibly EPS-water interaction. This same band was found to be important in another study of yogurt at the beginning of fermentation, which authors connected to the presence of still liquid milk [41]. However, the same study showed that the entire fermentation process could be monitored at this fixed wavelength—the changes in absorbance accurately reflected the fermentation process. The band at 1455 nm could also be assigned to the first overtone of O–H stretching of histidine [128], an amino-acid produced by LAB, a usual component of high-protein food, and commonly found in yogurt in high quantity.

Taken together, these interpretations suggest that the third stage of fermentation was characterized by several complex phenomena; each of them connected to the state of water in yogurt. Yogurt is one of the most interesting dairy products, in the sense that it has the highest water content, and yet it is solid. The ability to retain the water or its water holding capacity is one of the most important quality parameters of yogurt. Separation of the liquid phase, called syneresis, is one of the most undesirable occurrences in yogurt production, and it occurs when the gel protein matrix of yogurt is unable to firmly retain water. This ability to retain water is a result of the microstructure of yogurt, where the protein network consists of short branched chains of casein micelles that can be fortified additionally with compounds such as EPS, producing, in the end, a microstructure resembling sponge with very small pores. The higher density of this network and the smaller the size of the pores, the stronger will be the water-holding capacity of the yogurt; consequently, the viscosity and hardness are increased.

In the third stage, when the yogurt is produced, the interpretation of spectra suggests that the structure of the yogurt was set, with the majority of water being strongly bonded in ice-like structures (>1500 nm absorbance) [57], with minor fractions being held less firmly owing to the interaction with proteins and physically adsorbed. It is possible that bands 1420 nm and 1453 nm indicate some extent of bacterial inactivation and that there is a possibility of syneresis since the water-absorbing at these two bands, especially the second one, being only physi-adsorbed, can easily escape from the yogurt matrix.

In summary, the NIR spectra in stage three of fermentation reflect the state of water confinement in the protein-EPS matrix of yogurt that can be connected with mechanical and textural properties and even indicate the future behavior in terms of possible syneresis.

The interpretation of spectral features in this work, found to be descriptive of the fermentation process, relies mainly on the depiction of different water species-free water molecules, water solvation shells, protonated water, bound water, and adsorbed water. In agreement with previous aquaphotomics study on bacteria, the same absorbance bands of small protonated water clusters, free water molecules, weakly hydrogen-bonded water, and water-protein interactions were found to characterize the probiotic bacteria workings [59]. While some of the found bands may be attributed to the functional groups of other biomolecules, having in mind the large difference in their quantity compared to the amount of water in yogurt, it is presumed that the rest of the molecules in the media influence and coordinate the water and lead to changes in the water bands. While the properties of yogurt, such as viscosity, hardness, and others, are usually described in terms of coagulation phenomena and gel formation, the key element in the basis of these processes is binding (confining) water and preventing free movement of water [110]. Future studies that include the measurements of other conventional parameters (in addition to pH, like it was performed in this one) could reveal their relationship with the spectral features of yogurt during fermentation, better harnessing the power of NIRS technology.

As the texture of the yogurt is decisive for the final quality, further research into connecting spectral features of water in yogurt to textural properties could provide a new way to determine when the fermentation is completed, and desirable texture attained.

## 5. Conclusions

In this study, NIR spectroscopy was proposed for non-invasive automated control of yogurt production and aquaphotomics for a better understanding of the lactic acid bacteria fermentation process. The results showed that NIR spectra successfully captured both chemical and physical variations that occur during the fermentation process, suggesting that NIRS can be successfully applied to the monitoring of the process.

The interpretation of the obtained results showed the informational richness of the NIR spectra and further confirmed the exceptional role of the water in the sensing process since almost all the bands which tell the story about the ongoing fermentation process originated either from the water itself or from the interaction of water with the milk/yogurt components. Results obtained herein provide a proof of concept and excellent potential of NIRS for non-destructive, non-invasive monitoring of yogurt making. Provided that validation of experimental results is achieved on a larger number of samples in the replicated experiments, following the protocol described here, a valuable implementation in industrial settings for automated controlling of fermentation could be expected. With continuous real-time control, the NIRS can provide feedback about the state of the ongoing process and signal the need for corrective actions (for example, halting the process, adding fortifiers, etc.).

The information that NIRS can offer by far surpasses the one obtained by only measuring the pH, a single process parameter. Since the technology and its usage are very simple and low-cost, it could have a significant impact on world-wide yogurt production practices ensuring high and consistent product quality and production efficiency. Moreover, since the method is basically universal, it may also be applied in the same facilities for other, similar purposes.

## Figures and Tables

**Figure 1 sensors-21-00177-f001:**
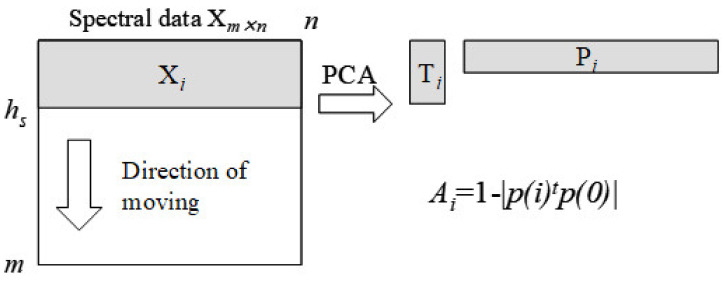
Schematic picture of the algorithm of moving window principal component analysis (MWPCA): X*_i_* is *i*th submatrix of spectral data X*_m_*_× *n,*_ and *h_s_* is the time-window size of X*_i_*. P*_i_* and T*_i_* indicate the principal components and scores of X*_i,_* respectively. *A**_i_* is a dissimilarity index to detect a change between angles of the first principal component in P*_i_*, *p*(*i*), and its reference, *p*(0).

**Figure 2 sensors-21-00177-f002:**
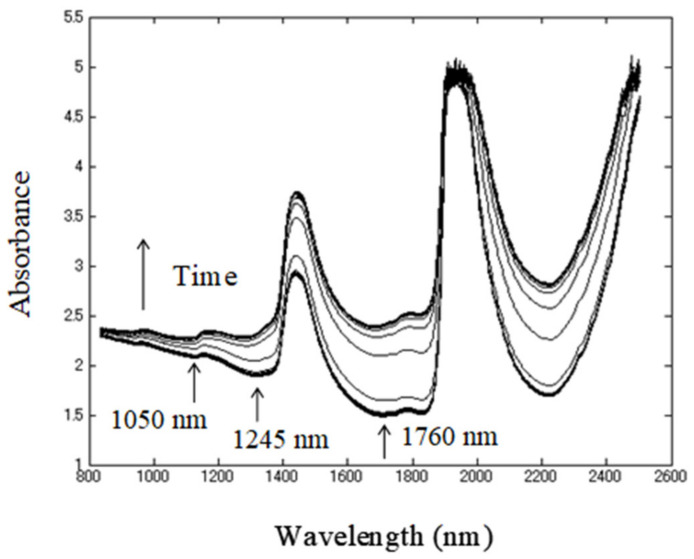
Raw near-infrared (NIR) absorbance spectra of yogurt acquired during 8 h fermentation.

**Figure 3 sensors-21-00177-f003:**
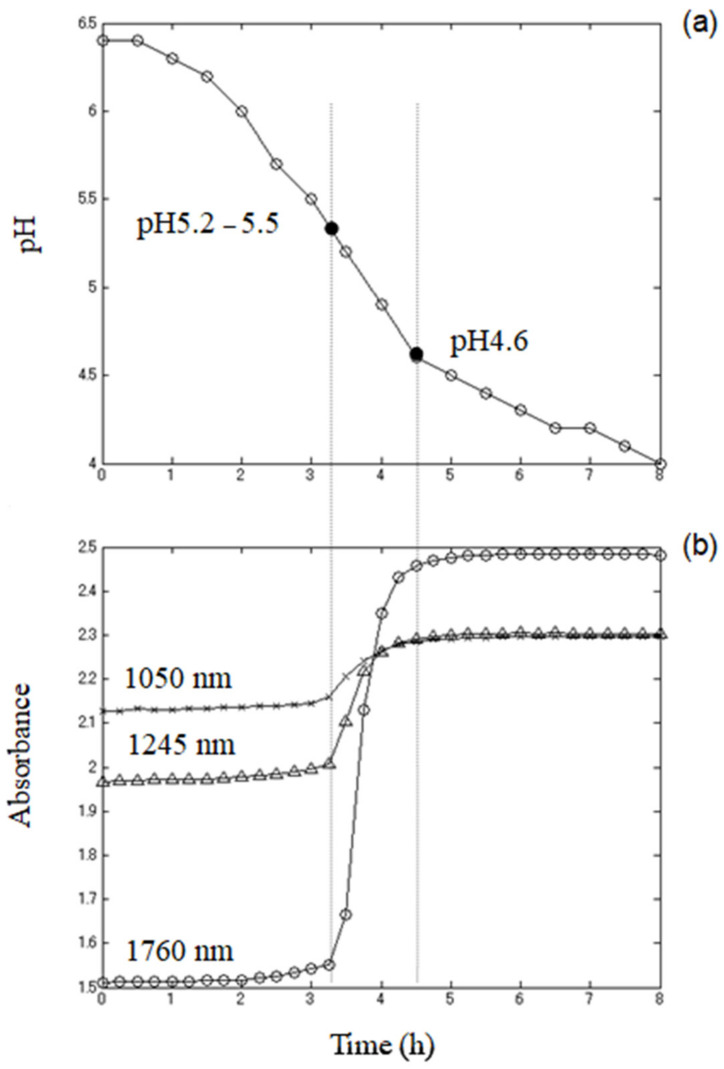
(**a**) pH profile of yogurt during fermentation (**a**,**b**) absorbance profiles at 1050 nm (×), 1245 nm (Δ), and 1760 nm (○) with coagulation of yogurt.

**Figure 4 sensors-21-00177-f004:**
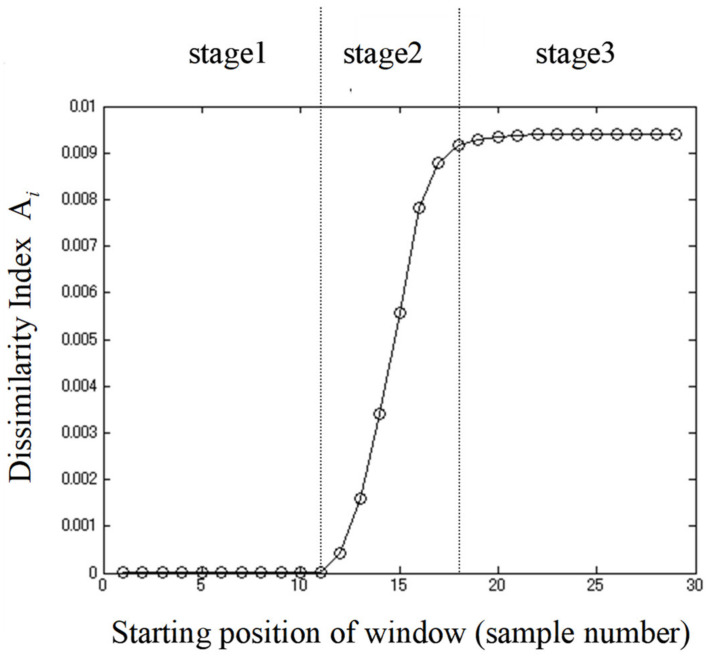
Dissimilarity Index *A_i_* based on loading variations of yogurt fermentation time (X-axis indicates the starting position of the window).

**Figure 5 sensors-21-00177-f005:**
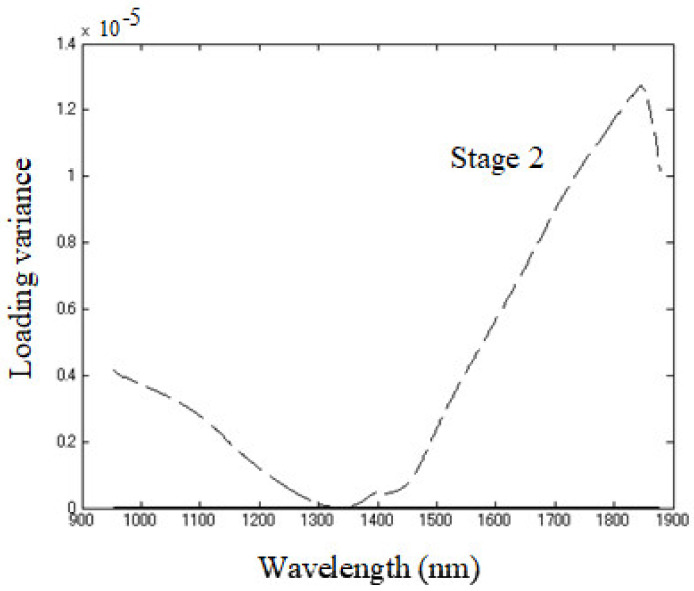
Loading variance for stage two (the period between 3 h 15 min and 4 h 30 min after the beginning of fermentation).

**Figure 6 sensors-21-00177-f006:**
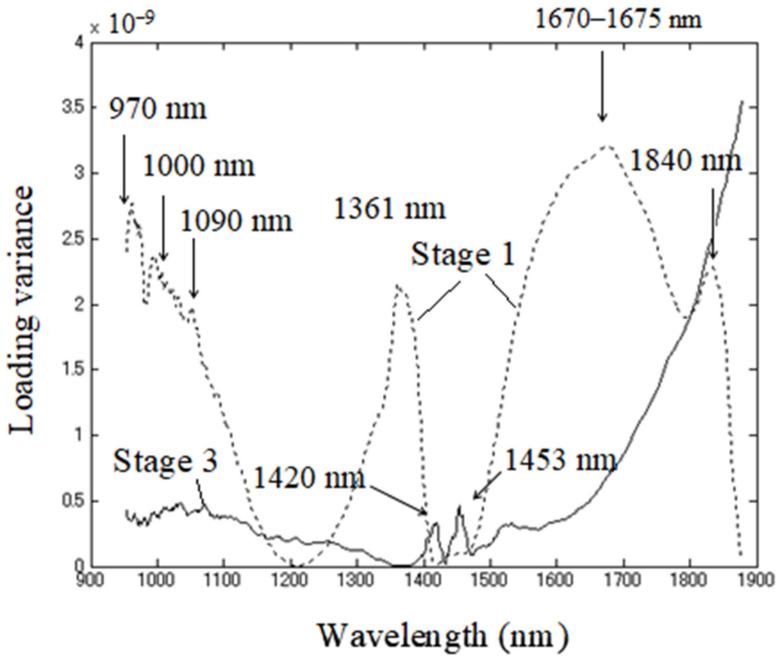
Loading variance for stage one (the period from the beginning of fermentation—0 h until 3 h and 15 min of fermentation) and stage three (from 4 h 30 min after the beginning of fermentation until the end of monitoring, 8h later).

## Data Availability

The data presented in this study are available on request from the corresponding author.

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
