# Peer review of "Real-Time Monitoring of Yogurt Fermentation Process by Aquaphotomics Near-Infrared Spectroscopy"

_sensors, 2020, doi:10.3390/s21010177_

Round 1

Reviewer 1 Report

Original Submission

Recommendation: Accept after minor revision

Comments to Author:

Manuscript number sensors-1049886

Title: Real-Time Monitoring of Yogurt Fermentation Process by Aquaphotomics Near-Infrared Spectroscopy

Authors: Jelena Muncan, Kyoko Tei, Roumiana Tsenkova

Overview and general recommendation: The paper reports the study on the real-time monitoring of the Yogurt Fermentation Process by the means of Aquaphotomics and the use of a near-infrared spectroscopy instrument for a non-invasive automated control. The UHT sterilized milk was inoculated with Bulgarian yogurt and placed into a quartz cuvette and test-tubes for capturing NIR spectra and pH in a period of 8 hours. The study deals with an interesting topic regarding the use of Moving Window Principal Component Analysis (MWPCA) to detect absorbance bands most involved in variations of the loading and caused by the LAB fermentation. Aquaphotomics is a well-established and known procedure, and NIR is one of the techniques already used in yogurt fermentation and quality evaluation. Notwithstanding this, the proposed extended use of NIRs and Aquaphotomics is a novelty and interesting topic. The manuscript is well written, it contains a captivating and well-motivated introduction. The materials and methods section is widespread and describes the applied materials together with a correct presentation of the statistical analysis used. To perform the MWPCA, the authors used a windows size = 5. Is the choice of the number supported by any reason? The study might be interpreted as explorative and descriptive of the likely application of NIR and Aquaphotomics techniques applied for yogurt fermentation monitoring, otherwise, in my opinion, there is a lack in the experimental and modelling construction in terms of evaluation of milk variability and processing conditions. Did the authors evaluate if the fermentation in a cuvette might be representative of the industrial process? Can the use of a single milk sample be representative of the real variability? In their opinion, could the presented curve shape for the loading variance in the three-stages and the dissimilarity index, and the found selected three-wavelength, differ significantly with the use of different milk or conditions? The measurement for pH is very interesting in terms of the evaluation of endpoint for fermentation. The authors measured the pH trend against time and compared it to the spectral shape of the found three most important wavelengths absorbances, by the same way it would be interesting to measure and investigate other rheological and conventional quality parameters changes in the fermentation process. Despite this, under the reported conditions, the research is valid and interesting in the results. The discussion presentation is well organized and discussed with other researchers’ results which increase the value of the chapter. Conclusions are satisfactory and supported by results and remarkable for the suggested implementation.

Verify the use of the acronyms MWPCA (lines 21-298- 307) or MPCA (lines 155-156-158-161-165-177-257-270) for moving window principal component.

For these reasons, I would recommend the Editor to accept the manuscript after minor revision.

Reviewer 2 Report

The paper title Real-Time Monitoring of Yogurt Fermentation Process by Aquaphotomics Near-Infrared Spectroscopy, provide interesting results and approach, and could be very useful for the industry.

In my opinion the paper could be accepted after some minor correction.

In conclusion the authors write: “… it could have a significant impact on world-wide yogurt production practices ensuring high and consistent product quality and production efficiency” I think this is true, but a sentence concerning the fact that before its application in the industry it will be necessary a validation process and a replication with a larger number of samples, is need.

In material and methods is not clear how many samples was measured and if the results express the average of them.

Is important explain why fermentation temperature is 38 °C.

For the equation in line 173 is important identify also what “t” means.

Title “3.2. Monitoring of chemical characteristics in fermentation of yogurt” must be changed, if fact the authors only measure the pH and the spectral analysis is related to functional groups but, in this case, is not a direct chemical analysis.
